# Progress and stagnation: A comparative study of gender representation in Chinese language textbooks for junior high school between 2001 edition and 2023 edition

Xia Yang[1], Yu Sun[2], Hu He[1,3]*

1 School of Teacher Education, Anqing Normal University, Anqing, China, 2 Academy of China's Rule-of-Law, East China University of Political Science and Law, Shanghai, China, 3 Institute of Educational Neuroscience, Anqing Normal University, Anqing, China

* hehu@aqnu.edu.cn

## Abstract

Gender bias in textbooks is a global problem. Gender development stands as a pivotal concern among junior high school students, underscoring the paramount importance of scrutinizing gender portrayals within textbooks. Chinese language textbooks, utilized by around 200 million students, have garnered scant attention in English literature concerning its embedded gender ideology. This essay delves into a comparative analysis of gender representations in Chinese Language textbooks published by the People's Education Press in 2001 and 2023, revealing little advancements. The evolution of these textbooks has witnessed a surge in the number of female authors and an expansion of gender-centric themes, fostering a more diverse array of female roles. However, a discernible disparity persists in the frequency of male and female portrayals, with males continuing to dominate the narrative, even amidst a slight decline in the female proportion. Furthermore, the spectrum of gender roles and professions remains limited, perpetuating the age-old gender binary of "men as providers, women as homemakers." Notably, societal stereotypes of both genders linger, impeding the depiction of nuanced and multifaceted identities. In summary, this study found that both sets of Chinese language textbooks implicitly contain gender bias, with little progress made towards gender equality, and even has stagnated or declined.

## Introduction

Junior high school students stand at the threshold of puberty, a pivotal juncture where their sexual awareness begins to blossom, marking a crucial milestone in their psychological development. This emerging awareness holds immense significance, shaping their self-identity, relationships, and overall emotional well-being. Gender

**Data availability statement:** All relevant data are within the manuscript, and there is no need to supplement them through Supplementary Information or any URLs/accession numbers/DOIs. The supplementary materials merely provide detailed information about some figure.

**Funding:** Support from the key research program of Anqing Normal University "Research on Determining Primary School Chinese Language Teaching Content Based on Text Style (SK202208ZD)", the National Education Science Planning Ministry of Education Youth Project "The Classification of Interdisciplinary Programs in Education in China and America: An Event History Model. (EIA 220526)", and the project of Talent of Education Law in Shanghai (2023JYFXR077) is gratefully acknowledged.

**Competing interests:** The authors have declared that no competing interests exist.

roles refer to the behavioral patterns assigned to individuals based on their gender by society, guiding junior high school students into becoming the kind of men or women they will be. Gender psychologists indicate that the development of adolescents' gender roles results from the construction of gender-related information and feedback in their environment, with social construction being the main influencing factor in the development of gender roles. School is the primary social environment for adolescent students, and textbooks are one of the main learning materials for students to study, which may profoundly impact their social cognition and personality development due to their implicit gender ideology [1]. This bias in gender representation profoundly affects the growth and development of girls. Some scholars argue that gender inequality leads to gender differences in brain structure [2], prevalence of depression [1], academic achievement [3], and more risk of female suicide [4]. To overcome these issues, pursuing gender equality has become a global consensus.

While gender equality has become a global consensus, further research has found that gender bias in textbooks is a global problem. Gender inequality is prevalent in textbooks in Africa [5,6], America [7,8], Australia [9], Asia [7,9], Europe [7,10,11] and the United Kingdom [12]. Islam and Asadullah conducted a quantitative content analysis on language textbooks from Malaysia, Indonesia, Pakistan, and Bangladesh, they found that the cultural differences among these countries led to quantitative distinctions in gender representations in the textbooks. In a country, the more unequal the gender gap, the less female roles are portrayed in its textbooks, and gender stereotypes are more severe [13]. Dong and Li revealed the prevalence of gender inequality in textbooks across different countries and cultural backgrounds through a meta-analysis of global textbook research, although the degree varies [14]. The Chinese language textbook, used by around 200 million people worldwide, has garnered scant attention in English literature concerning its embedded gender ideology. It is worth exploring and studying whether it also harbors gender inequality.

Gender inequality in the textbook is important because language (in textbook) and gender are mutually constructive. The gender concept can be shaped in students' understanding of the textbook. Weatherall highlighted that "at discursive level, language about women (and men) and women (and men) speaking are both aspects of one process – the social construction of gender, which is the constructionist perspective." [15]. Litosseliti demonstrated that learners may internalize the gender values embedded in textbooks, perceiving them as credible and authoritative sources [16]. In this context, both visual and verbal representations of gender in textbooks require careful consideration, as noted by Widodo, Fang and Elyas [17]. To achieve this, applied linguists have increasingly called for a critical perspective in the design and utilization of language textbooks, emphasizing the need for materials that are sensitive to gender values [18].

In the context of gender equality becoming a global consensus, China has also actively responded to the call for gender equality. Since the Republic of China's Nanjing government explicitly recognized gender equality in its constitution in 1947, governments of different periods in China have affirmed the principle of gender equality from the legal level. Since 1978, the Constitution, Civil Law, Criminal Law, and

Administrative Law of the People's Republic of China have upheld gender equality as a fundamental national policy. Since 2000, the status of Chinese women has seen tremendous improvement. In universities, the proportion of female students exceeds 50%, and in hospitals, the percentage of female doctors is over 46% [19]. Furthermore, women account for 35% to 38% of corporate board directors and supervisors, and constitute more than 50% of grassroots neighborhood committee members [20]. However, previous research on Chinese Language textbooks in China has revealed that they continue to be permeated with male-centeredness and stereotypes [21]. This is evident in the limited presence of female characters, with women more often depicted in nurturing or caregiving roles. Furthermore, textbooks rarely mention female doctors or female officials [22].

Chinese Language is one of the core curricula that junior high school students need to learn, exerting a significant impact on their gender awareness development. The Chinese language textbook, utilized by around 200 million students, has garnered scant attention in English literature concerning its embedded gender ideology. Since 2000, the 2001 edition of the Chinese language textbook served as the first set of textbooks for the 21st century, while the 2023 edition represents the later version. Do these textbooks reflect China's social progress and address the longstanding gender biases that have existed? We compared and analyzed the differences in gender roles between the 2001 and 2023 versions of Chinese language textbooks used in China over the past two decades, to evaluate whether the 2023 version had made progress in gender equality over time.

## Literature review

The importance of revealing gender ideology in school textbooks lies in the fact that implicit gender representations in subject curricula carry authoritative social knowledge and values, which deeply influence students' cognition and behavior. Numerous previous studies have revealed gender inequality in textbooks from different domains.

### Number of male and female authors

The statistical analysis of the number of male and female authors can reflect the degree of gender equality in textbooks. This dimension is mainly adopted by Chinese researchers due to the selective nature of Chinese language textbooks, which select famous authors' works as the content of textbooks. For example, MA Guoyi's study of the junior high school Chinese Language textbooks published by PEP in 1998 found that there were a total of 162 male authors among 182 authors, accounting for 89.01% of the total number of authors [23]. Similarly, HUANG Qiaomei pointed out that in the 1991 edition of the Hong Kong and Macao junior high school Chinese Language textbooks, there were a total of 107 authors, of which 100 were male, accounting for 93.5% of the total, while there were only 7 female authors, accounting for 6.5% [24].

### Frequency of male and female roles

The statistics on the frequency of gender roles in textbooks is a key way to examine gender bias and encompass information from both text and images. Some researches have shown that the proportion of female characters in textbooks is less than that of males. For instance, Jackie F. K. Lee and Peter Collins's comparative study of Hong Kong secondary English textbooks published in the late 1980s/early 1990s found that female characters account for 37.2% while the proportion of male characters is 62.8% in earlier textbooks [25]. In addition, in Jackie F.K. Lee's study of Hong Kong textbooks published in 1995, the frequencies of female and male characters were 47.2% and 52.8%. In Chinese Language textbooks, the proportion of male characters is also higher than that of females [26]. For example, WANG Wei found that male roles account for 63.74%, while female roles account for 36.26% in Chinese primary school textbooks. The above studies have shown the insufficient presence of women [27].

However, there was an exception where the percentage of female characters was higher than that of male characters. Jackie F. K. Lee and Peter Collins found that female roles account for 51.02%, while male roles account for 48.98% in

Hong Kong secondary English textbooks published in 2008 [25]. Nevertheless, merely increasing the number of women without changing their representation cannot solve the problem of gender bias.

### Number of male and female protagonists

The number of male and female protagonists in textbooks is also an important indicator for measuring gender equality. Countless researchers have documented that textbooks consistently place male figures at the center of narratives, while women are more frequently pushed to the periphery, relegated to supporting roles for men. Miroiu's and Biemmi's studies revealed that female characters are underrepresented as protagonists [28,29]. Jehle and colleagues uncovered that among 2233 protagonists, only 874 were female, constituting a mere 39%. Their analysis revealed a distinct gender imbalance: females appeared significantly less frequently than males as protagonists within mathematics and language textbooks from Germany, Italy, Lithuania, Netherlands, and Romania [30].

### Gender stereotypes

The stereotypical portrayal of men and women is another evidence of gender inequality in textbooks. For example, Alrabaa's study of 28 textbooks from grades 8–12 in Syria revealed that 84% were filled by males and were filled 16% by females in a total of 463 occupations. Men mainly were engaged as presidents/kings, soldiers, professionals, and farmers, while a small number of women were engaged in traditional jobs (such as nurse) but the majority were in economically dependent domestic roles [31]. WANG Wei found that activities were gender-specific in Chinese primary school textbooks. Dealing with diplomatic tasks, fighting, hunting, horse racing, and so on were identified largely as male activities while teaching, shopping, praying, feeding, nurturing, and so on were seen as female activities. Therefore, men have a wider range of activities than women [27].

Correspondingly, this factor shaped different personality traits between males and females. Boys were brave, independent, ambitious, and sometimes naughty, while girls were passive, obedient, tidy, and cooperative [32]. Blumberg discovered stereotypes of both genders' occupational and household roles that overwhelmingly underplay women' s rising worldly importance [7]. Jackie F. K. Lee and Peter Collins showed women were weaker and more passive than men in Australian and Hong Kong English language textbooks [25].

Chinese scholars have also carried out a lot of investigation on gender ideology in Chinese junior high school textbooks. MA Guoyi and his colleagues' study on the 1998 edition of People's Education Press' junior high school Chinese Language textbooks found that there were phenomena of neglecting women in the selection of texts, mainly manifesting in two aspects: Firstly, the outdated and traditional female portrayal, lacking modern representations of strong and independent women; secondly, the male-dominated selection of texts, with a preponderance of works by male authors and female characters portrayed according to male fantasy, making women subordinates to men [23].

The present study selected the 2001 and 2023 editions of junior high school Chinese Language textbooks as samples. And analyzing the frequency of female and male appearances, family roles and social/professional roles, we explored whether the content of textbooks reflected the requirements of Chinese laws and regulations for gender equality, and whether the images of women in textbooks were consistent with the socio-economic status of women in real society in the past two decades, aimed to evaluate whether textbooks had made progress in gender equality.

## Methods and issues

### Textbooks

Chinese language textbooks were used in the present study because Chinese Language is a compulsory subject in junior high school in China, and students have to spend about 6 hours per week learning Chinese Language. This study selected two editions of junior high school Chinese Language textbooks (a total of 12 books) published by People's

Education Press (PEP) in 2001 and 2023. The People's Education Press is a large professional publishing enterprise affiliated with the Ministry of Education of the People's Republic of China founded in 1950, the press has been devoted to editing, publishing, and distributing educational publications, making outstanding contributions to China's education industry. The 2001 edition which was already out-of-print at the time of the study followed the principle of "one syllabus, multiple editions", however, it was still the mainstream textbook in China at that time, and the range of usage was also the widest. While the 2023 textbook edition followed the principle of "one syllabus, one book", which was revised from the 2017 edition and was the only Chinese Language textbook in PRC. The two editions of junior high school textbooks each have 6 volumes (a total of 12 volumes), which are used in Grade 7, Grade 8, and Grade 9, see Table 1.

Our research focuses on analyzing publicly published school textbooks, with no copyright disputes involved. The images include portraits of the text's author, photographs of the main characters in the text, and illustrations of fictional characters from the novel. None of the individuals depicted in the pictures are participants. The whole research does not involve any human or animal participant, nor does it involve collecting or harming any living organisms, and the Ethics Committee considered that an informed consent procedure was not required, therefore, we do not possess any authorization legal documents or ethical statements.

## Quantitative study

To explore the representation of gender in Chinese Language textbooks, a Chinese Language teaching expert with 15 years of teaching experience conducted quantitative coding on the textbook. The quantitative analysis results which covered the frequency and percentage of specific coding elements were obtained through content analysis of textbooks. The quantitative analysis is crucial to emphasize the inclusion and exclusion, visibility and invisibility of male and female roles in the textbooks.

Quantitative data was collected by encoding different gender descriptions in Chinese Language textbooks. Based on Lee and Collins' study of Australian English textbooks and HUANG, P and LIU, X' s study of English textbooks [25,33], this study presumed the analysis categories of textbooks as author gender, media (including text and picture), gender stereotypes (family role and social role), frequency of male and female roles, and number of male and female protagonists, see Table 2 for details.

## Qualitative study

Quantitative analysis is unable to reveal the gender representations that are concealed within the context, therefore, the present study employed an improved critical discourse analysis (CDA) for qualitative analysis. The qualitative analyst and the quantitative analyst are the same teaching expert.

**Table 1. Textbooks.**

| 2001 edition textbook | | | | | 2023 edition textbook | | | | |
|---|---|---|---|---|---|---|---|---|---|
| Vol. | Grade | Year | Page | ISBN | Vol. | Grade | Year | Page | ISBN |
| 1 | 7(1) | 2001 | 208 | 978-7-107-14630-0 | 1 | 7(1) | 2023 | 156 | 978-7-107-31244-1 |
| 2 | 7(2) | 2001 | 281 | 978-7-107-14904-0 | 2 | 7(2) | 2023 | 169 | 978-7-107-31488-9 |
| 3 | 8(1) | 2001 | 269 | 978-7-107-14905-4 | 3 | 8(1) | 2023 | 160 | 978-7-107-31931-0 |
| 4 | 8(2) | 2009 | 268 | 978-7-107-22295-5 | 4 | 8(2) | 2023 | 140 | 978-7-107-32361-4 |
| 5 | 9(1) | 2003 | 268 | 978-7-107-16572-6 | 5 | 9(1) | 2023 | 149 | 978-7-107-32805-3 |
| 6 | 9(2) | 2009 | 244 | 978-7-107-22294-8 | 6 | 9(2) | 2023 | 148 | 978-7-107-33121-3 |
| Total | 3 | | 1538 | | Total | 3 | | 922 | |

**Table 2. The coding framework.**

| Domains | Categories | criteria | Codes |
|---|---|---|---|
| Number of male and female authors | Author gender | Appearance of the female and male authors | Female; Male |
| Frequency of male and female roles | Male and female roles | Appearance of female and male roles in the written text. *e.g.,HE Manz i* (何满子) *grew up by his grandmother's side, wanting the stars in the sky, and his grandmother quickly moved the ladder to pick them*. (1 male and 1 female character) | Female; Male |
| | Visual representation | Note the number of men and women in each picture, including the front cover, to determine if the picture depicts only male role, only female role, or a mix of both. If the picture contains only female role, count female one; if it includes only male role, count male one; and if there are both male and female roles in the picture, and both male and female roles are counted as one each. | Female; Male |
| Number of male and female protagonists | Male and female protagonists | Note the number of male and female protagonists in the written text and picture. The judgment criteria for the protagonist in the text are as follows: (1) The roles mentioned in the title (e.g., in "Flying in the Sky - Record of Diving Girl LV Wei Winning the Championship", LV Wei is the protagonist.), (2) The central figures in the plot (e.g., in "The Steps", the father is the protagonist.), (3) The main participants in conflicts and contradictions (e.g., in "Face -Changing", *Shuishangpiao* and *Dogchild* are the protagonists.), (4) The roles described in a large number of words in the text. (e.g., in "The Beauty of Radiance", Marie Curie and Pierre Curie are the protagonists.) The judgment criteria for the protagonist in the picture are as follows: (1) The picture of the protagonist in the text, (2) The roles located in the center of the picture. For instance, in the illustration titled "FAN Jin Passing the Provincial Examination and Achieving Success" featured in the 2023 textbook (2023 edition, Volume 5, p. 129), FAN Jin occupies the central position within the image, thereby establishing him as the protagonist. (3) The roles with relatively large image sizes. For instance,in the illustration "Pear Blossoms along the Post Road" from the 2023 textbook (2023 edition, Volume 2, p. 99), the girl positioned at the forefront is depicted with the largest scale, thus designating her as the protagonist. | Female; Male |
| Gender stereotypes | Familial roles | Identify the familial roles in the written text. If a role has multiple identities, calculate them in addition. (e.g., In the text "A Walk", "I" am both a son and a father, counted as 1 son and 1 father.) | Mother, father, daughter, son, wife, husband, etc. |
| | Social/occupational roles | Identify the social/occupational roles in the written text. If a role has multiple identities, calculate them in addition (e.g., In the text "From the Hundred-Plant Garden to the Sanwei Study)", "I" am both a writer and a student, counted as 1 writer and 1 student.). If a figure is both a social role and a family role, calculate them in addition. (e.g., In the text "The Beauty of Radiance", Pierre Curie is described as both a scientist and a husband, counted as 1scientist and 1 husband.) | Scientist, leader, teacher, doctor, etc. |

CDA can be seen as a method [34], as a theory or perspective [35]. It is derived from critical theory established originally by Marxists and the Frankfurt school. It is considered discourse analysis with a critical attitude towards social problems, especially focusing on how discourse helps power reproduction. There are three improved approaches in CDA: Fairclough's dialectical-relational approach, Wodak's discourse-historical approach, and van Dijk's socio-cognitive approach. The dialectical-relational approach assumes that language and society mutually constitute and determine

each other, with this mutual relationship dialectical. The discourse-historical approach emphasizes the importance of historical background and contextualizes discourse against the background. The socio-cognitive approach introduces psychological models into linguistic analysis [35]. This study mainly applies Fairclough's dialectical-relational approach which matches social analysis on gender. Fairclough proposes three steps of critical discourse analysis: description, interpretation, and explanation [36]. The method of discourse analysis includes linguistic description of the language text, interpretation of the relationship between the (productive and interpretative) discursive processes and the text, and explanation of the relationship between the discursive processes and the social processes [36]. In gender studies, CDA aims to examine the seemingly "normal" gender representation in textbooks, by considering language and social relationships, as well as deconstructing discourse to reveal its hidden power operating in social contexts, make "implicit" and " unspoken" transparent and visible [35].

## Result

### The result of quantitative study

**Author gender.** Although men had higher author representation in both editions of textbooks, there was an increase slightly female authors in the 2023 edition. The frequency of female authors has increased from 8.65% in the previous editions to 9.09% in the 2023 edition, see Table 3.

Incorporating the works of female writers into textbooks is a respect and emphasis on women's literary creation and expression rights. In ancient China, literary creation was long dominated by men, and female writers like LI Qingzhao, who enjoyed a great reputation in ancient Chinese history, were also rare. Female writing was suppressed and not recognized. It was not until modern China that a large number of female writers began to emerge. In the 2001 version of the textbook, 23 works by female writers are included, covering 18 female writers, because some articles are written by the same author. In the 2023 edition of the textbook, the number of female works remained at 23, and the number of female writers climbed to 22. This change signifies slight progress in gender equality within the unified edition of textbooks, as newly included female writers in the unified edition textbook are XIAO Hong, MAO Ning, QIU Jin, LIN Huiyin, BI Shumin, J.K. Rowling, Rachel Carson, and others.

In summary, although the 2023 edition textbooks are still dominated by male authors, compared to the previous set of textbooks, this textbook has shown slight progress in gender equality in terms of the proportion and presentation of female authors.

**Female and male roles.** This section covers three aspects: (1) the frequency of male and female characters in the written text of the two editions of textbooks, (2) the presentation of gender roles in the illustrations of the two editions of textbooks, and (3) the appearances of male and female protagonists in the two editions of textbooks.

(1) frequency of male and female roles

The result revealed that the frequency of female characters in the text of the 2023 edition has decreased from 27.65% to 25.42%, a decrease of 2.23%, see Table 4.

Table 3. Number of male and female authors.

| Edition | Frequency | | | Percentage | |
|---|---|---|---|---|---|
| | Men | Women | total | Proportion of men(%) | Proportion of women(%) |
| 2001 edition | 243 | 23 | 266 | 91.35 | 8.65 |
| 2023 edition | 230 | 23 | 253 | 90.91 | 9.09 |

**Table 4. Frequency of female roles in texts.**

| Edition | Frequency | | | Percentage | |
|---|---|---|---|---|---|
| | Men | Women | total | Proportion of men(%) | Proportion of women(%) |
| **2001 edition** | 212 | 81 | 293 | 72.35 | 27.65 |
| **2023 edition** | 264 | 90 | 354 | 74.58 | 25.42 |

**Table 5. Frequency of female roles in pictures.**

| Edition | Frequency | | | Percentage | |
|---|---|---|---|---|---|
| | Men | Women | total | Proportion of men(%) | Proportion of women(%) |
| **2001 edition** | 222 | 62 | 284 | 78.17 | 21.83 |
| **2023 edition** | 175 | 41 | 216 | 81.02 | 18.98 |

**Table 6. (a) Men and female protagonists in pictures. (b) Male and female protagonists in texts.**

| Edition | Frequency | | | Percentage | |
|---|---|---|---|---|---|
| | Men | Women | total | Proportion of men(%) | Proportion of women(%) |
| *(a) Men and female protagonists in pictures.* | | | | | |
| **2001 edition** | 102 | 24 | 126 | 80.95 | 19.05 |
| **2023 edition** | 53 | 8 | 61 | 86.89 | 13.11 |
| *(b) Male and female protagonists in texts.* | | | | | |
| **2001 edition** | 99 | 32 | 131 | 75.57 | 24.43 |
| **2023 edition** | 107 | 34 | 141 | 75.89 | 24.11 |

(2) presentation of gender roles in the illustrations

Compared to the previous edition, the 2023 edition textbooks are more unequal between genders. Similar to the previous edition, the number of male characters still holds a significant advantage in the 2023 edition textbooks. In the images of the two editions of textbooks, the 2001 edition textbook had 222 male characters and 62 female characters, accounting for 21.83% of the total. The number of male and female characters in the 2023 edition textbooks is 175 and 41, with female characters accounting for only 18.98% of the total, a decrease of 2.85% compared to the previous set of textbooks, see Table 5.

(3) the appearances of male and female protagonists in the two editions of textbooks

After investigation, it is found that the number and proportion of female protagonists in 2023 edition textbooks are at a lower level in both illustrations and texts compared to 2001 edition textbooks. Firstly, from the illustrations, in the 2001 edition textbook, the proportion of male protagonists is as high as 80.95%, while the proportion of female protagonists is only 19.05% (see Table 6a). In contrast, there are 53 male protagonists and 8 female protagonists in the 2023 edition textbooks, the proportion of male protagonists has further increased to 86.89%, while the proportion of female protagonists has decreased to 13.11% (see Table 6a). Comparing the two sets of textbooks, the gap in the proportion of male and female protagonists has widened, and the proportion of female protagonists has decreased by 5.94%. Secondly, from the text of the textbooks, the gender role distribution in the 2001 version of the textbooks shows a total of 99 male protagonists and 32 female protagonists, with the proportion of male protagonists is 75.57%, while the proportion of female protagonists is 24.43%. In contrast, in the 2023 version textbooks, the number of male protagonists

has increased to 107, while the number of female protagonists is 34, with the proportion of male protagonists has increased to 75.89%, while the proportion of female protagonists has correspondingly decreased to 24.11% (see Table 6b). By comparing the two versions of the textbook, it is observed that the gap in the proportion of male and female protagonists in the 2023 edition textbooks has slightly widened, while the proportion of female protagonists has slightly decreased.

To conclude this section, the proportion of female-to-male appearances is less in recent textbooks than in earlier ones, both visually and textually. These findings reveal that women not only have been under-represented in the two editions of Chinese Language textbooks in junior high school textbooks over the past two decades, but also the imbalance has increased in the more recent books. Based on the above analysis, it can be concluded that the 2023 edition junior high school Chinese textbooks have not shown improvement consistent with the progress of relevant laws and regulations on gender equality issues, but have instead shown a regression phenomenon.

**Gender stereotypes.** This section covers two aspects: (1) the frequency of male and female roles in the family, and (2) the frequency of male and female in social/ occupational roles.

(1) family roles

According to a survey, in the 2001 edition, male roles account for 60.00% in the family domain while female roles only account for 40.00%, which reflects gender imbalance. In contrast, the proportion of male characters has decreased to 48.58% in the 2023 edition, while the proportion of female characters has increased to 51.42% (see Table 7), which may reflect some progress in gender equality. Consistent with Jackie F.K. Lee's investigation of Hong Kong primary school ELT textbooks [25], the result suggests that the representation of both genders is more balanced in the recent edition, indicating the progress in gender equity in school textbooks over the past few decades.

(2) social/occupational roles

This study includes an analysis of the social/professional roles played by women and men in textual content. In the number of women occupying social/occupational roles over the past two decades, we observe a modest increase in female representation within the social domains, rising from 11.76% in the 2001 edition to 13.11% in the 2023 edition (see Table 8). It is also worth noting that although traditional social/professional stereotypes associated with women and men still exist, there are occasional portrayals of women as athletes, soldiers, scientists, astronauts, and men as tailors or servants.

**Table 7. Distribution of familial roles.**

| Textbooks of 2001 edition | | | | Textbooks of 2023 edition | | | |
|---|---|---|---|---|---|---|---|
| **Familial Roles** | **Men** | **Familial roles** | **Women** | **Familial Roles** | **Men** | **Familial roles** | **Women** |
| son | 60 | mother | 38 | son | 30 | mother | 28 |
| father | 48 | wife | 25 | father | 24 | wife | 23 |
| husband | 24 | grandmother | 13 | husband | 17 | daughter | 15 |
| grandson | 12 | daughter | 12 | grandfather | 5 | grandmother | 9 |
| grandfather | 9 | younger sister | 6 | older brother | 5 | granddaughter | 9 |
| younger brother | 9 | elder sister | 5 | uncle | 5 | younger sister | 7 |
| older brother | 8 | daughter-in-law | 5 | nephew | 4 | elder sister | 4 |
| other | 13 | other | 18 | other | 13 | other | 14 |
| **Total** | 183 (60.00%) | | 122 (40.00%) | | 103 (48.58%) | | 109 (51.42%) |

**Table 8. Social/occupational roles.**

| Edition | Frequency | | | Percentage | |
|---------|-----------|---------|-------|------------|---------|
| | Men | Women | Total | Proportion of Men(%) | Proportion of Women(%) |
| 2001 edition | 120 | 16 | 136 | 88.24 | 11.76 |
| 2023 edition | 179 | 27 | 206 | 86.89 | 13.11 |

### The result of qualitative study

Quantitative analysis has revealed gender biases in some Chinese textbooks for junior middle school students. However, the frequency and percentage indicators used in quantitative analysis are unable to uncover the gender representations that are hidden within the context. Therefore, this study will proceed to employ Critical Discourse Analysis (CDA) for qualitative research, to analyze the gender representations concealed in the context.

**The social identity and writing style of female authors.** After a thorough qualitative analysis of the identity of female writers in 2023 edition textbooks, we further found that textbooks also reflected progressive features in other aspects. Firstly, the diversity of authors' identity prosperously grows compared to the previous version. The 2001 version covers 18 female authors of which 15 are Chinese and three are Caucasian. Concerning occupation, only one is a scientist while all the other 17 are litterateurs of which 12 are essayists. In contrast, the 2023 version contains five Caucasian female authors. Concerning ethnicity, the 2023 version removes eight Chinese essayists and adds new authors with multiple occupations as journalists, playwrights, marine biologists or scientists, art scholars, publishing house editors, modern architects, social activists, and fantasy novelists. In the new edition of the textbook, the identity of female authors is diversified, showing the breakthrough of contemporary women to traditional gender roles and their contributions to social civilization and progress. Secondly, with the increase in the number of female writers and the diversification of their identities, the themes and presentation techniques of textbook content have also shown corresponding diversity. Compared to the previous edition, where most female authors' writing styles tend to be emotional, lyrical, and focused on describing a life, the recent edition not only reflected the sad and troublesome side of female authors but also rose above mere emotional expression to address ideological and aesthetic pursuits.

For example, BI Shumin builds three small houses for "love and hate", "career", and "self" (2023 edition, Volume 5, p. 40, "Three Little Houses of the Spirit"). Charlotte Bronte depicts Jane Eyre's resilient, self-respecting, and independent spiritual character (2023 edition, Volume 6, p. 143, "Jane Eyre"). MA Lihua depicts the magnificent scenery of the snowy plateau, reflecting the author's delicate and rich emotions (2023 edition, Volume4, p. 100, "At the Source of the Yangtze River in Kelantan Winter"). QIU Jin expresses her concern for the national crisis, as well as her enthusiasm and determination to save the country (2023 edition, Volume 6, p. 60, "Man Jiang Hong·A Brief Stay in Jinghua"). Rachel Carson reflects the author's reflection on human conquest and transformation of nature, which has had a significant impact on the cause of environmental protection (2023 edition, Volume 3, p. 134, "Silent Spring"). These selected works cover a wide range of themes, including the growth process of women, individual spiritual fields, magnificent mountain scenery, patriotism, and environmental issues.

**Female and male representation.** A qualitative study and CDA of gender representation in textbooks has revealed instances of gender inequality. For example, in the story "Grandma Liu Visits the Grand View Garden" (2023 edition, Volume 5, p. 137), there are 10 important female characters. Although Chinese artists have created various illustrations for this story, none of them are included in the textbooks. The only female scientist character in the text, Marie Curie, is depicted in a photograph with her husband, Pierre Curie, rather than a solo photo of herself. On the contrary, in a junior high school Chemistry textbook published by PEP for use in China, to give chemist Antoine Lavoisier a separate photo, the editor cut out his wife from the couple's photo (2001 edition, Volume 5, p. 17). Why couldn't the editor of the Chinese textbook provide Mary with a personal portrait photo separately? A study found that when female scientists are depicted

in textbook illustrations, girls tend to perform better in science classes [37]. Furthermore, in some illustrations, female images were often marginalized, existing only as backgrounds for male images, or not presented. For example, in "My Uncle Jules", the illustrations of the three female characters are small in size and blurry (2023 edition, Volume 5, p. 81). In "FAN Jin passes the Provincial Examination and becomes a Successful Candidate", the image of FAN Jin's wife is placed at the edge and back of the screen, and her gaze is directed towards FAN Jin (2023 edition, Volume 5, p. 129). These female characters are limited to the background or spectators of male characters. And in the 2001 edition textbooks, these two illustrations are limited to male images and did not include female characters.

Further qualitative analysis and CDA reveals the avoidance and neglect of gender topic in the 2023 edition of textbooks, for example: (1) the deletion of the text "Face-Changing" (as below) that reveals gender bias, and (2) the selective introduction of QIU Jin.

*Shuishangpiao: Don't be too tired, come here and take a rest. (Holding the child in his arms)*

*Dogchild: (Hugging) Grandpa, you're so kind.*

*Shuishangpiao: I'm kind because I have high hopes for you. First, I fear breaking the family bloodline. Second, I fear the loss of our ancestral skill. That's why I've taken you as my grandson. I sincerely treat you, teach you the art of changing faces, and hope you will carry on my lineage.*

*Shuishangpiao: Let me teach you a few sentences first, you need to remember them. A family heirloom skill, passed down within but not outside the family, passed down to sons but not daughters. To pass on to a daughter is to betray the ancestors, and one will be struck by lightning from heaven.*

*Dogchild (singing)*

*I am a little girl! There is joy and sorrow, a little girl meets a good old man. I dare not speak my worries aloud, fearing that my grandfather will abandon me!* (2001 edition, Volume 6, p. 131–132)

*Shuishangpiao: I won' t sell you, but I won't keep you either. Boys are treasures, girls are weeds—damn it all, I want the treasure not the weed! Here is some traveling money and provisions for you. Take them and make your own way in life.*

*Dogchild (singing)*

*A thousand regrets, ten thousand regrets, wrong to be born a girl through heaven's bet. I'm more diligent than a boy, can do heavy work and carry loads. I'll wash your bedding in the river, chop dry wood to cook your meals... It is worthwhile to have a filial granddaughter!*

*Shuishangpiao (singing)*

*Where's the family bloodline? I'm ashamed before the ancestral shrine. Only by raising sons can the lineage be continued. Daughters marry out, disasters entwine. Valuing males over females since ancient times!* (2001 edition, Volume 6, p. 135–137)

Fairclough's steps of CDA involve the linguistic description of the text, interpreting the relationship between texts and discourse practices, and explaining the connection between discursive practices and socio-cultural practices [36].

Firstly, from the perspective of the text, many language expressions used in the text reflect the ancient Chinese society's "preference for sons over daughters" ideology and the oppression of women by male-dominated society. The text originates from the 2001 edition of the textbook titled "Face-Changing," featuring a dialogue between *Shuishangpiao* and *Dogchild*. *Shuishangpiao* is a single folk artist with extraordinary skills. *Dogchild* is a poor orphan who had been repeatedly rejected and sold because of her female identity, so she disguised herself as a boy. *Dogchild* is bought from a human

trafficker by *Shuishangpiao* to be his grandson and apprentice, so as to inherit his face-changing art. But when *Dogchild's* true identity as a girl is discovered, *Shuishangpiao* plans to kick her out. From the point of view of roles, they are grand-father and grandson as well as master and apprentice. From the point of view of gender, it is a conflict between male and female. In terms of the name, only a girl can be adopted if she takes a boy's name (*Dogchild* is a boy's name), which reflects the miserable fate and low status of the girl. From the perspective of gender descriptive metaphors, women are called "broom star" and "weed", men are called "treasure", and even boys' urine is valuable medicine ("*boy's urine, mixed with cloth ash, an ancestral formula, to reduce swelling and detoxify*", 2001 edition, Volume 6, p. 134). At the same time, the art of face-changing can only be passed on to boys, not girls, and if it is passed on to girls, they will suffer ("*To pass on to a daughter is to betray the ancestors, and one will be struck by lightning from heaven.*"), which reflects gender stereo-types and social norms ("*Valuing males over females since ancient times.*").

Secondly, it is discussed in terms of discourse practice. The author of the text, WEI Minglun, is a famous contem-porary playwright. He advocates women's liberation, opposes male hegemony, and creates some female images who dare to resist the idea of male superiority and female inferiority. Face-Changing is a work that deeply reflects on the traditional concept of "valuing men over women". Using the genre of script, the author presents the tragic fate of women in a patriarchal society through dialogue, singing and nursery rhymes in the dramatic conflict, and exposes the harm brought to women by gender contempt. For example, the affection ("*Holding the child in his arms*") and hope ("*I have high hopes for you*") for *Dogchild* are all based on the premise that *Dogchild* is a boy. Once it is discovered that *Dog-child* is a girl, *Shuishangpiao* cruelly drive her away ("*I won't keep you*"). Although *Dogchild* tries her best to be a boy ("*I'm more diligent than a boy, can do heavy work and carry loads. I'll wash your bedding in the river, chop dry wood to cook your meals.*"), she is still compelled to leave and nearly drowns and dies. This text, widely disseminated through Chinese textbooks, is a valuable resource. It offers teachers the opportunity to guide students in breaking gender ste-reotypes and promoting gender equality. This indicates that the editors of the 2001 edition textbook regarded gender equality as important.

Thirdly, from the perspective of social and cultural practice. On the one hand, the text reveals the traditional patriar-chal ideology of China and the historical fact of gender power inequality: men have held power, possessed discourse, and have been dominant within the family and society. In terms of the family, only men are entitled to inherit the family's bloodline, and children typically take on the father's surname. This can be found in the words of *Shuishangpiao* in the text ("*Only by raising sons can the lineage be continued.*"). From a social perspective, men traditionally have the privilege of acquiring knowledge and attaining high social status. Knowledge, identity, status, and power are inherited and per-petuated among men. This reinforces the oppression and marginalization of women. On the other hand, men have two dimensional of "patriarchal strategies" to protect their interests [38]. The first is that men oppress women by preventing them from accessing the necessary productive resources of the economy (e.g., the knowledge or skills of "Face-changing" in the text). The second is that women's labor is perceived as insignificant compared to men's, trapping them in a closed cage [38]. For instance, even though *Shuishangpiao* lives alone, *Dogchild* is still driven away despite her willingness to take on household chores like laundry and cooking. Women's labor as caregivers is devalued and thus further oppressed. This is not a problem of individual men, but rather the systematic exclusion and devaluation of women by the male group as a whole, which forces women to endure their disadvantaged position in reproduction [38]. Moreover, women have inter-nalized the gender ideology and power structure of male supremacy and female inferiority under such oppression. For example, *Dogchild* saves another boy for *Shuishangpiao* because she agrees with the view that only men can be heirs. It is evident that the abolition of patriarchy can be achieved not only by altering men's attitudes and gender consciousness but also by transforming women's. Gender equality can be realized by changing institutional and power structures in real-ity (such as the material basis, economic relations, knowledge acquisition, and the education system).

However, the 2023 edition of Chinese textbooks has omitted the text "Face-Changing" to circumvent the topic of gender equality, and no alternative texts with a similar theme have been provided. Social transformation, social structural change

and redistribution of power require the rewriting of textbooks to reflect the ideology of the new era. This reflects the mode of power operation during the process of textbook composition and approval. Therefore, the textbook itself is a product of power struggles. Textbooks realize the function of "cultural reproduction" of social power and gender ideology, reflecting the interests and values of men as a dominant group. The male-centered mainstream culture integrates other cultures, thereby obscuring the values of women as a disadvantaged group. At the same time, The "Discussion and Review" section after the text designs a discussion: "*The idea of favoring sons over daughters has always been deeply rooted in our country. Connect your observations and share your views on this issue with your classmates.*" (2001 edition, Volume 6, p. 138) This is a valuable opportunity for students to recognize gender bias, think about gender equality issues, and change gender attitudes. The omission of this text, which addresses patriarchal ideology, reflects the stagnation of the 2023 edition textbooks on gender equality.

In addition, it is worth mentioning that QIU Jin is not only an outstanding poet and martyr of the democratic revolution, but also an active advocate of Chinese feminism and women's studies. However, the textbook only emphasizes QIU Jin's identity as a poet and patriot, without introducing her achievements as a feminist in the field of feminism, which is incomplete for QIU Jin. Once again, it is seen that textbook editors avoided the topic of gender equality.

**Gender stereotypes.** Merely increasing the frequency of women without addressing the manner they are portrayed does not reveal the problem of gender bias [26]. The qualitative study of the types and nature of gender representation in family and society reveals that their portrayal is not free of gender stereotypes.

Firstly, we explore family roles. Men usually play the role of decision-makers and leaders in the domestic sphere, while women are assistants. The manner of mothers in textbooks is often depicted as a symbol of tolerance, demonstrating dedication, selflessness, kindness, loving, capable, diligent, warm, giving, and sacrificial. For instance, in "Autumn Nostalgia", the mother is described as follows:

"*My mother quietly hid out and listened to my movements in a place I couldn't see.*"

"*The chrysanthemums in Beihai have bloomed, let me push you to take a look.*" Her haggard face showed a pleading expression.

The last words before she fainted were: "*My son who is ill and my daughter who is still underage.*" (2023 edition, Volume 1, p. 20–21)

The text uses words such as "quietly," "hiding," and "pleading" to describe the mother (for example, "She quietly hid herself away" "pleading expression"), these descriptions of the mother's "silencing" convey the kindness, restraint, and patience of maternal love. Even when the mother was seriously ill and before she fell into a coma, her greatest concern was for her children (*"my son who is ill and my daughter who is still underage"*). The text does not fully romanticize the mother's sacrifice, for instance, details such as "*she was still spitting up blood in large mouthfuls*" (2023 edition, Volume 1, p. 21) and "*she often suffered from liver pain, tossing and turning all night long, unable to sleep,*" (2023 edition, Volume 1, p. 21) indicate the suppression and suffering behind the individual of the noble maternal role. These descriptions align with the societal conception of the perfect mother: a caregiver at the cost of self-sacrifice. A mother is only considered great and perfect in the act of selfless devotion. This reinforces the stereotype of the mother's image. Erich Fromm believes that the role of the mother is not only to provide children with a sense of security in life but also that motherly love should not hinder their growth or encourage dependency [39]. Instead, it should foster independence and eventual detachment. The image of mothers in textbooks are disproportionately cast as mere sources of emotional comfort. Harsh and visionary mothers who foster autonomy seldom appear.

In contrast, the father's image is characterized by strictness and authority. The father has high expectations and requirements for the children, providing spiritual encouragement and helping them grow and develop. For instance, the following conversation between father and son:

*"No, I can't."* I howl. *"It's too far, it's too hard, I can't do it."*

*"Listen to me,"* my father says. *"Don't think about how far it is. All you have to think about is taking one little step. You can do that. Look where I'm shining the light. Do you see that rock?"* The beam bounces around on a jutting outcrop just below the ledge. *"See it?"* he calls up. (2023 edition, Volume 1, p. 78)

The father in the text is strict, calm, and wise, he leads the son to overcome difficulties and fears, benefiting his son for life. In addition, the father in The Steps believed that the steps in front of the house were low and decided to raise them, which was supported by the mother (2023 edition, Volume 2, p. 67), and although dissatisfied, the wife still obeyed her husband in the child's marriage (2023 edition, Volume 6, p. 35). Finally, both male and female characters have cases of counter-stereotype, such as the father who climbed onto the platform to buy oranges for his son, demonstrating love and affection (2023 edition, Volume3, p. 75). The wife and grandmother with a high voice, good at cursing and fighting, she was proficient in life skills such as farming, rowing, fishing, childbirth, and medical treatment, demonstrating a strong practical spirit and aggressive style (2023 edition, Volume 6, p.33).

Secondly, concerning social roles, the investigation reveals that the occupational roles and personality traits of male and female roles reflect gender stereotypes. Men tend to focus more on their careers, engaging primarily in creative and intellectual occupations and playing a more significant role in the social fields. For example, in "The Declaration of Creativity" (2023 edition, Volume 6, p. 107), none of the 11 individuals mentioned as being creative are women. On the other hand, when it comes to recalling men, few textual content that emphasizes their fatherhood, while more texts focus on their diligence, career, or contributions, such as "DENG Jiaxian" (2023 edition, Volume 2, p. 2), "Speak and Do" (2023 edition, Volume 2, p. 9), "Memories of LU Xun" (2023 edition, Volume 2, p. 13), and "SUN Quan Encourages Learning" (2023 edition, Volume 2, p. 22).

## Discussion

### Progress and stagnation in textbooks: More female roles, but still men-centered

The purpose of this study is to investigate whether the rapid development of China's economy and society, along with the implementation of gender equality-related laws, has impacted the representation of gender in textbooks over the past two decades. The present study conducts a quantitative and qualitative analysis of numerous core indicators, and find that with the improvement of gender awareness in China. An increase in the representation of women in the recent edition has been reflected in several key dimensions, including the number of female authors, and their representation of family roles and social roles. The present study find the representation of men and women in the family field is gradually towards equality: men and women are depicted as fathers and mothers, husbands and wives, sons and daughters. The finding is consonant with some previous studies [26,40,41]. The results of this study indicate that gender representation in textbooks is improving, especially in the family domain where the gender role ratio tends to be balanced, which is an essential first step in promoting gender equality in textbooks [9].

The presentation of the author, protagonist, family role, and social role shows a clear male-centered tendency in both sets of textbooks. As Caroline Criado Perez pointed out, when women are seen as a secondary gender class, they often become exceptionally conspicuous [42]. However, when they need to be valued and included in statistics, they are often overlooked [42]. This phenomenon reflects injustice and prejudice against women. Therefore, it is evident that the editors of junior high school Chinese textbooks have not given sufficient attention to the existence of women. Some scholars believe that being ignored is not conducive to developing self-esteem. Self-concept theory emphasizes that self-esteem plays an important role in career preferences and career choices. People with low self-esteem, especially women, may have difficulty matching themselves with their professional roles. Low self-esteem can limit women's pursuit of traditionally male-dominated careers and reduce career aspirations [43].

In summary, there remains considerable room for enhancement in the portrayal of female figures within the unified edition Chinese Language textbooks utilized in junior high schools. Meanwhile, the recognition of women's social status also faces severe challenges, which urgently need to be taken seriously and addressed by textbook writers.

## Stereotypes are laden with textbook: Women are still considered the second sex

The examination reveals that the occupational characteristics and personality traits of male and female roles reflect gender stereotypes. For example, in the political field, the majority of roles are occupied by men, while female roles are very scarce. Even in seemingly ordinary leadership roles, such as captains, ship owners, shopkeepers, and restaurant owners, the individuals depicted are predominantly male. Similarly, in the medical field, female doctors are often excluded from representation. These portrayals do not reflect current realities. It is consistent with Sunderland's finding that women appear more frequently in low-status professions, the only social profession where the number of women exceeds that of men is servant, women are more often involved in caregiving work, which is still a reflection of female stereotypes [44]. The passivity of "feminine" women in essence is a trait that developed when they were young [45].

The textbooks in this study ignore social changes and the evolution of times, continuing to adhere to the ancient patriarchal gender norms. In these norms, men are portrayed as pioneers, creators, heroes, and warriors, while women are seen as caregivers, assistants, and passive figures. The patriarchal gender norms and representation have led to gender biases and stereotypes. As carriers of ideology, textbooks inadvertently impart these biases and stereotypes to children from an early age. This is in line with Beauvoir's viewpoint: "Women are not naturally formed from their social gender. It is the pervasive gender culture that permeates every subtlety of social life that monitors, shapes, and carves women into the second sex." [46]

In addition to the phenomenon of "women second", the tendency of "male first" is also worth noting. The number of female roles in the social field is far less than that of male roles, while they are repeatedly emphasized in the family field. This comparison leads to women losing their independence and freedom, and subsequently becoming subordinate to men. This suppresses women's potential for development and limits their personal growth. Textbooks do not fully present women's social contributions and values, which is undoubtedly unfair and biased. Merely increasing women's appearance without addressing the manner they are portrayed does not solve the problem of gender bias [26]. This conservative and traditional gender representation has not improved in the past 20 years but has instead become increasingly severe. "We need to allow more women to prove their value as human beings, rather than allowing society to evaluate women based on their ability to raise offspring." [45].

## Underrepresentation of women: Gender ideology lags behind the times

After the establishment of the People's Republic of China, the protection of women's status and rights became a highly concerned and actively promoted issue by the government. Women enjoy in terms of political rights, economic and labor security rights, cultural and educational rights, and marriage and family rights, providing a solid legal foundation for safeguarding women's legitimate rights and interests and promoting gender equality. According to The Global Gender Gap Report [47] and the Final Statistical Monitoring Report on the Implementation of the Outline for the Development of Chinese Women [20], more than 1,000 women serve as deputies to the National People's Congress and members of the Chinese People's Political Consultative Conference, and women account for 35% to 38% of corporate board directors and supervisors, more than 50% of grassroots neighborhood committee members. The number of female doctors has shown a trend of surpassing male doctors [19]. Statistical data shows that Chinese women are at the forefront of economic participation and opportunities in the world, accounting for about 45% of China's total human resources [20]. Chinese women have played a wide range of roles in socio-economic development and have demonstrated their enormous influence in social life.

However, the status of women and legal requirements are not reflected in Chinese Language textbooks. The concept of gender equality in textbooks has not improved synchronously with the development of social economy and civilization progress, and even shows a trend of degradation [30,48]. Consistent with previous research, the present survey shows that men still appeared more often than women in the visual and textual forms of the 2023 edition. Compared with the 2001 edition textbooks, the 2023 edition textbooks show a trend of degeneration in some dimensions, including the proportions of female illustrations, female roles, and female main characters. The traditional gender role division model of "men leading the outside and women leading the inside" in 2023 edition textbooks has not broken through the limitations of the traditional framework, and there has been no significant improvement compared to the People's Education Press textbooks 20 years ago. Engels pointed out that "in any society, the degree of women's liberation is the natural measure of universal liberation." However, it is worth noting that the 2023 edition of the textbook is reducing the number of texts on gender issues.

This study also aimed to evaluate whether the presentation of women has reflected social realities in China. Despite data proving that Chinese women have significant participation and positive contributions in various fields such as the economy, politics, and education, the depiction and frequency of female images in textbook written texts and illustrations are still seriously insufficient. Moreover, it shows a trend of decline in the frequency of female characters in text, the proportion of women in the picture and the appearances of protagonists in the picture and the written text of the 2023 edition textbooks. What is even more noteworthy is that Chinese Language textbooks are avoiding discussions on gender issues by omitting texts and guided materials.

## Conclusion

The present study has provided insights into gender construction in junior high school Chinese Language textbooks published in the 21st century, and finds slihgt progress in promoting gender equality in textbook content by textbook editors, the textbook editors strive to avoid gender bias and recognize the importance of creating gender-equal teaching materials. However, the current study reveals that Chinese Language textbooks implicitly contain gender bias. Although contemporary China has seen a large number of women participating in social labor, the influence of patriarchy rooted in traditional Chinese culture remains strong. The concepts reflected in textbooks still mirror this long-standing patriarchal ideology, while the principle of gender equality upheld by Chinese laws and policies has not sufficiently influenced the content of Chinese Language textbooks. The path to gender equality in Chinese Language textbooks in China remains a long one, due to the underrepresentation of women and the persistence of stereotypes.

This study suggests that education in China, like in other countries, may also be out of step with the times and contain gender biases. This suspicion and concern are solely prompted by an examination of two versions of Chinese junior high school Chinese Language textbooks. The present study primarily attributes gender awareness in textbooks to editorial stance, yet alternative explanations remain unexplored. Future research should explore other aspects, such as textbooks for different grades or subjects, teaching methods employed by teachers, campus culture, and socio-cultural environments. The coding process was independently completed by one teaching expert for 12 books, so subjectivity is inevitable. Future research could be carried out by more people to code the data and provide error indicators.

Gender inequality has become a global consensus that hinders human development and the progress of nations. Textbooks, which occupy an absolutely important position in the field of education, need to actively promote gender equality [33]. In terms of reassessing the value of women, textbooks should keep pace with social reality, or at least not fall behind reality. Chinese Language textbooks should reflect the spirit of the times and the principle of legal gender equality. Language textbooks should discard the long-standing patriarchal ideology, especially those traditionally stereotyped as "unsuitable for their gender". These efforts are necessary to promote gender equality in textbooks [45].

Gender bias in Chinese textbooks is very detrimental to the development of female middle school students. As widely implemented in UNESCO's guidelines for gender-sensitive teaching materials, China also urgently needs to develop clearer guiding principles for textbook compilers and education practitioners to promote gender equality. This guiding

principle may include strategies such as balancing male and female roles, shaping social roles that reflect social reality, and reducing descriptions of gender stereotypes, etc.

## Author contributions

**Data curation:** Xia Yang.

**Formal analysis:** Xia Yang, Yu Sun, Hu He.

**Funding acquisition:** Xia Yang.

**Investigation:** Xia Yang.

**Methodology:** Yu Sun, Hu He.

**Project administration:** Hu He.

**Resources:** Xia Yang.

**Supervision:** Hu He.

**Writing – original draft:** Xia Yang.

**Writing – review & editing:** Yu Sun, Hu He.

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
