## [Decision Letter · Decision Letter 0]

17 Mar 2025

Dear Dr. He,

Both reviews had positive comments on your manuscript, but further revisions are required, in order clarify the method used and the discussion the manuscript belongs to.

We look forward to receiving your revised manuscript.

Kind regards,

Rafael Galvão de Almeida, PhD.

Academic Editor

PLOS ONE

Journal Requirements:

For additional information about PLOS ONE ethical requirements for human subjects research, please refer to http://journals.plos.org/plosone/s/submission-guidelines#loc-human-subjects-research .

“Support from the key research program of Anqing Normal University “Research on Determining Primary School Chinese Language Teaching Content Based on Text Style�SK202208ZD�”, the National Education Science Planning Ministry of Education Youth Project “The Classification of Interdisciplinary Programs in Education in China and America: An Event History Model. (EIA 220526)”, and the project of Talent of Education Law in Shanghai (2023JYFXR077) is gratefully acknowledged.”

5. PLOS ONE publication criteria require that the data presented in the manuscript must support the conclusions drawn, and submissions will be rejected if the interpretation of results is unjustified or inappropriate, so authors should avoid overstating their conclusions. We noted the following potentially overstated and/or unclear statements in the Results section that require revision:

Section 4.1.1:  "This change signifies significant progress in gender equality"

It is unclear whether this statement is supported by the presented results, as the observed change appears to be small and statistical significance is not reported.

Section 4.1.3:  "In contrast, the proportion of male characters has decreased to 48.58% in the 2023 edition, while the proportion of female characters has increased to 51.42% (see table 7), reflecting progress in gender equality."

We believe this may be an overstatement and would recommend rephrasing “reflecting progress in gender equality." to "which may reflect some progress in gender equality." or similar.

Section 4.2.1: "Firstly, the identities of female writers are more diverse, such as journalists, marine biologists, playwrights, theorists, scientists, publishing house editors, modern architects, fantasy novelists, etc. among the newly added female writers"

This statement is unclear, as it is unknown what this diversity is being compared against. For example, comparison groups could include male writers of the same textbook editions or female writers of previous editions.

Section 4.2.1:  "They demonstrate the independent spirit of modern women, break through the constraints of traditional roles, show the potential for self-expression and realization, and contribute to social change and civilization development".

This statement does not appear to be supported by the data presented.

Section 4.2.2 - "Regrettably, the 2023 edition of Chinese textbooks has omitted the text 'Face Changing' to circumvent the topic of gender equality, and no alternative texts with a similar theme have been provided".

This statement does not appear to be supported by citation, and more supported information regarding omitted text is needed.

Finally, PLOS ONE requires that the language in submitted articles must be clear, correct, and unambiguous. In this case, the manuscript requires editing for English grammar and usage as well as for typographical errors.

Reviewers' comments:

Reviewer's Responses to Questions

**Comments to the Author**

1. Is the manuscript technically sound, and do the data support the conclusions?

Reviewer #1: Yes

Reviewer #2: Partly

2. Has the statistical analysis been performed appropriately and rigorously?

Reviewer #1: Yes

Reviewer #2: N/A

3. Have the authors made all data underlying the findings in their manuscript fully available?

Reviewer #1: Yes

Reviewer #2: Yes

4. Is the manuscript presented in an intelligible fashion and written in standard English?

Reviewer #1: Yes

Reviewer #2: Yes

Reviewer #1: [Title of the Manuscript]Progress and Stagnation: A Comparative Study of Gender Representation in Chinese

Textbooks for Junior High School between 2001 Edition and 2023 Edition

[Manuscript ID] PONE-D-25-05629

[Overall Evaluation]

This manuscript discusses the use of frequency, expression, illustrations, and other methods to enhance the potential gender inequality in middle school Chinese language textbooks. By elevating the image and perspective of women in the textbooks, it aims to raise awareness of gender equality among young people and achieve future gender equality in society.

At present, in the research of Chinese textbooks, there are already analytical literature on character images and roles in Chinese texts, with more emphasis on the educational impact of positive typical characters in texts on students' growth; In the study of character illustrations in textbooks, more attention is paid to whether illustrations affect students' concentration, while there is currently little research on the impact of gender on characters in Chinese textbooks.

Advantages: 1. This manuscript provides a sensitive perspective on the analysis of the number, roles, professions, and other aspects related to women in Chinese and other language textbooks. It shows that the frequency of women's appearance in textbooks is not high, which also reflects a global issue of gender discrimination against women in life, career, personal values, and other aspects. This perspective is very valuable.

2. Whether the language is clear and the evaluation of research methods is appropriate, the data analysis is accurate, the results are reliable, and the charts are clear.

Improvement: There are some areas that need improvement before the manuscript can be considered for publication.

1. Current achievements and research directions related to Chinese language textbooks.

2. Adding constructive suggestions on improving the social status of women in textbooks to the theoretical framework can enhance the comprehensiveness and developability of the article.

[Conclusion]

In conclusion, this manuscript has the potential to make a valuable contribution to the field. With the suggested revisions, it could be suitable for publication in Plos One.

[Reviewer Name] Jiawei Shi

[Reviewer Affiliation] HUANGHE S&T UNIVERSITY

[Date of Review]2025.3.1

Reviewer #2: Dear Editor(s),

Thank you for the opportunity to review the manuscript entitled “Progress and Stagnation: A Comparative Study of Gender Representation in Chinese Textbooks for Junior High School between 2001 Edition and 2023 Edition.” The study addresses an important and timely issue, contributing valuable insights into gender representation in educational materials over time.

The article employs a well-structured mixed-methods approach, combining quantitative content analysis with qualitative critical discourse analysis. The quantitative analysis is particularly rigorous, offering a systematic breakdown of key indicators such as the frequency of male and female authors, characters, and protagonists. The inclusion of specific data points, such as the increase in female authors from 8.30% in 2001 to 9.09% in 2023, strengthens the credibility of the findings. Additionally, the qualitative analysis provides depth by uncovering implicit gender biases, such as the portrayal of Marie Curie alongside her husband rather than independently. This dual-method approach enhances the study’s comprehensiveness and allows for a nuanced understanding of gender representation trends.

The study is highly relevant within both local and global contexts. Gender representation in textbooks remains a significant issue worldwide, and the manuscript effectively situates its findings within broader international discourse by referencing studies from multiple regions. The focus on Chinese language textbooks, used by approximately 200 million students, fills a critical gap in the literature and offers insights that have direct implications for educational policy. The discussion on policy implications, particularly in light of China’s national commitment to gender equality, is particularly compelling. The study provides a balanced perspective, highlighting both progress and persistent challenges in gender representation over two decades.

The manuscript effectively identifies positive developments, such as the increasing number of female authors and the improved gender balance in family roles. However, it also underscores concerning trends, including the continued underrepresentation of women in social and professional spheres and the persistence of gender stereotypes. The discussion of regression in female protagonist representation, from 19.05% in 2001 to 13.11% in 2023, is especially noteworthy. The study’s critical engagement with these issues adds depth to the analysis and highlights the need for continued reform.

Despite these strengths, the methodology section requires further elaboration to enhance the study’s transparency and reproducibility. The manuscript does not provide sufficient detail on the coding process, inter-coder reliability, or statistical tests used in the quantitative analysis. Similarly, the description of the critical discourse analysis approach lacks specificity. The authors mention an “improved CDA,” but it is unclear what modifications were made and how they were implemented.

The manuscript exhibits significant repetition, particularly in the Results and Discussion sections. Key points, such as the underrepresentation of women and the persistence of gender stereotypes, are reiterated multiple times without adding new insights. Additionally, the discussion of regression in gender representation is presented in multiple sections but would benefit from a more integrated analysis.

While the study references existing literature, the engagement with theoretical frameworks and global perspectives on gender representation in textbooks could be strengthened. The manuscript cites key scholars such as Sunderland and Beauvoir but does not fully incorporate theoretical discussions on how gender stereotypes influence students’ career aspirations and self-perception. Furthermore, drawing on global practices, such as UNESCO’s guidelines for gender-sensitive teaching materials, would provide a stronger foundation for the study’s policy recommendations.

The manuscript’s readability is hindered by long, complex sentences and occasional grammatical errors. For example, sentences such as “The present study conducted a quantitative and qualitative analysis of numerous core indicators, and found that with the improvement of gender awareness in China, an increase in the representation of women in the recent edition has been reflected in several key dimensions” could be revised for conciseness and readability.

The study would benefit from a more in-depth exploration of its practical implications for educational policy and practice. While the findings highlight gaps in gender representation, the manuscript does not provide concrete recommendations for textbook authors, policymakers, or educators. Additionally, the omission of texts addressing gender equality is noted, but the manuscript does not fully explore how this might influence students’ perceptions or what corrective actions could be taken.

Finally, the manuscript would be strengthened by a discussion of counterarguments and study limitations. The analysis attributes gender disparities in textbooks primarily to editorial bias but does not consider alternative explanations, such as broader societal trends or historical constraints on available materials. Additionally, the study focuses on only two editions of Chinese textbooks, limiting its ability to capture long-term trends comprehensively. Acknowledging these limitations and discussing potential biases in the coding and analysis process would enhance the study’s credibility and scholarly contribution.

Overall, the manuscript provides a valuable contribution to the study of gender representation in educational materials. Some revisions would enhance its overall quality and impact. I appreciate the opportunity to review this work and look forward to seeing its continued development.

**Do you want your identity to be public for this peer review?** For information about this choice, including consent withdrawal, please see our Privacy Policy

Reviewer #1: No

Reviewer #2: **Yes: ** Xiaoming Tian

---

## [Author Response · Author response to Decision Letter 1]

16 May 2025

Response to Reviewers

1.Please ensure that your manuscript meets PLOS ONE's style requirements, including those for file naming.

Response: HAVE DONE.

2.Please provide additional details regarding participant consent. In the ethics statement in the Methods and online submission information, please ensure that you have specified what type you obtained (for instance, written or verbal, and if verbal, how it was documented and witnessed). If your study included minors, state whether you obtained consent from parents or guardians. If the need for consent was waived by the ethics committee, please include this information.

Response:Our research focuses on analyzing publicly published school textbooks, with no copyright disputes involved. It does not involve any human or animal subjects, nor does it involve collecting or harming any living organisms.

“Support from the key research program of Anqing Normal University “Research on Determining Primary School Chinese Language Teaching Content Based on Text Style�SK202208ZD�”, the National Education Science Planning Ministry of Education Youth Project “The Classification of Interdisciplinary Programs in Education in China and America: An Event History Model. (EIA 220526)”, and the project of Talent of Education Law in Shanghai (2023JYFXR077) is gratefully acknowledged.” Please state what role the funders took in the study. If the funders had no role, please state: "The funders had no role in study design, data collection and analysis, decision to publish, or preparation of the manuscript."

Response: The funders had no role in study design, data collection and analysis, decision to publish, or preparation of the manuscript. We have further stated this in cover letter and manuscript.

4.We note that your Data Availability Statement is currently as follows: All relevant data are within the manuscript and its Supporting Information files.

Response: All required data and raw data are provided in the text.

5.PLOS ONE publication criteria require that the data presented in the manuscript must support the conclusions drawn, and submissions will be rejected if the interpretation of results is unjustified or inappropriate, so authors should avoid overstating their conclusions. We noted the following potentially overstated and/or unclear statements in the Results section that require revision:

Section 4.1.1:"This change signifies significant progress in gender equality"

It is unclear whether this statement is supported by the presented results, as the observed change appears to be small and statistical significance is not reported.

Response: Thank you to the reviewers for their careful review and constructive criticism. We have changed this sentence to “This change signifies slight progress in gender equality”. See blue words in line 256. Thank you again to the reviewers.

Section 4.1.3: "In contrast, the proportion of male characters has decreased to 48.58% in the 2023 edition, while the proportion of female characters has increased to 51.42% (see table 7), reflecting progress in gender equality."

We believe this may be an overstatement and would recommend rephrasing “reflecting progress in gender equality." to "which may reflect some progress in gender equality." or similar.

Response: Thank you to the reviewers for their careful review and constructive criticism. We have changed this sentence to "which may reflect some progress in gender equality." see blue words line 324. Thank you again to the reviewers.

Section 4.2.1: "Firstly, the identities of female writers are more diverse, such as journalists, marine biologists, playwrights, theorists, scientists, publishing house editors, modern architects, fantasy novelists, etc. among the newly added female writers"

This statement is unclear, as it is unknown what this diversity is being compared against. For example, comparison groups could include male writers of the same textbook editions or female writers of previous editions.

Response: Thank you to the reviewers for their careful review and constructive criticism. We added detailed data and compared the identities of the female authors of the two versions of the textbook, see blue words from line 350 to line 357. Thank you again to the reviewers.

Section 4.2.1: "They demonstrate the independent spirit of modern women, break through the constraints of traditional roles, show the potential for self-expression and realization, and contribute to social change and civilization development".

This statement does not appear to be supported by the data presented.

Response: Thank you to the reviewers for their careful review and constructive criticism. This section is an explanation of the diversity of female authors in the new edition of the textbook. It may be overinterpreted. We have revised this section to read as follows, “In the new edition of the textbook, the identity of female authors is diversified, showing the breakthrough of contemporary women to traditional gender roles and their contributions to social civilization and progress”,see blue words from line 357 to line 360. Thank you again to the reviewers.

Section 4.2.2 - "Regrettably, the 2023 edition of Chinese textbooks has omitted the text 'Face Changing' to circumvent the topic of gender equality, and no alternative texts with a similar theme have been provided".

This statement does not appear to be supported by citation, and more supported information regarding omitted text is needed.

Response: Thank you to the reviewers for their careful review and constructive criticism. We have greatly expanded the quotation and discussion of the "face change" in the text, see blue words from line 408 to line 434. Thank you again to the reviewers.

Finally, PLOS ONE requires that the language in submitted articles must be clear, correct, and unambiguous. In this case, the manuscript requires editing for English grammar and usage as well as for typographical errors.

Response: HAVE DONE.

Reviewer #1:

This manuscript discusses the use of frequency, expression, illustrations, and other methods to enhance the potential gender inequality in middle school Chinese language textbooks. By elevating the image and perspective of women in the textbooks, it aims to raise awareness of gender equality among young people and achieve future gender equality in society.

At present, in the research of Chinese textbooks, there are already analytical literature on character images and roles in Chinese texts, with more emphasis on the educational impact of positive typical characters in texts on students' growth; In the study of character illustrations in textbooks, more attention is paid to whether illustrations affect students' concentration, while there is currently little research on the impact of gender on characters in Chinese textbooks.

Advantages: 1. This manuscript provides a sensitive perspective on the analysis of the number, roles, professions, and other aspects related to women in Chinese and other language textbooks. It shows that the frequency of women's appearance in textbooks is not high, which also reflects a global issue of gender discrimination against women in life, career, personal values, and other aspects. This perspective is very valuable.

Response: Thank you to the reviewers for their positive comments on the value of this paper.

Improvement: There are some areas that need improvement before the manuscript can be considered for publication.

1.Current achievements and research directions related to Chinese language textbooks.

Response: Thank you to the reviewers for their careful review and constructive criticism. In section 2.1, we added references and analysis of the literature on gender issues in Chinese textbooks, see blue words from line 110 to line 116. Thank you again to the reviewers.

2. Adding constructive suggestions on improving the social status of women in textbooks to the theoretical framework can enhance the comprehensiveness and developability of the article.

Response: Thank you to the reviewers for their careful review and constructive criticism. In the last two paragraphs of the conclusion, we added discussions on strategies to address and improve gender bias in the textbook, see blue words from line 679 to line 691. Thank you again to the reviewers.

Reviewer #2:

The article employs a well-structured mixed-methods approach, combining quantitative content analysis with qualitative critical discourse analysis. The quantitative analysis is particularly rigorous, offering a systematic breakdown of key indicators such as the frequency of male and female authors, characters, and protagonists. The inclusion of specific data points, such as the increase in female authors from 8.30% in 2001 to 9.09% in 2023, strengthens the credibility of the findings. Additionally, the qualitative analysis provides depth by uncovering implicit gender biases, such as the portrayal of Marie Curie alongside her husband rather than independently. This dual-method approach enhances the study’s comprehensiveness and allows for a nuanced understanding of gender representation trends.

The study is highly relevant within both local and global contexts. Gender representation in textbooks remains a significant issue worldwide, and the manuscript effectively situates its findings within broader international discourse by referencing studies from multiple regions. The focus on Chinese language textbooks, used by approximately 200 million students, fills a critical gap in the literature and offers insights that have direct implications for educational policy. The discussion on policy implications, particularly in light of China’s national commitment to gender equality, is particularly compelling. The study provides a balanced perspective, highlighting both progress and persistent challenges in gender representation over two decades.

The manuscript effectively identifies positive developments, such as the increasing number of female authors and the improved gender balance in family roles. However, it also underscores concerning trends, including the continued underrepresentation of women in social and professional spheres and the persistence of gender stereotypes. The discussion of regression in female protagonist representation, from 19.05% in 2001 to 13.11% in 2023, is especially noteworthy. The study’s critical engagement with these issues adds depth to the analysis and highlights the need for continued reform.

Response: Thank you to the reviewers for their positive comments on the value of this paper.

Despite these strengths, the methodology section requires further elaboration to enhance the study’s transparency and reproducibility. The manuscript does not provide sufficient detail on the coding process, inter-coder reliability, or statistical tests used in the quantitative analysis.

Response: Thank you to the reviewers for their careful review and constructive criticism. To enhance the study’s transparency and reproducibility, We have revised the description of the encoding process,see “Quantitative study” and “Qualitative study” in paper. The coding process was independently calculated by the first author of the paper for 12 books for a long time, so errors are inevitable and it is impossible to estimate its inter-coder reliabiliby. We have stated this in the conclusion part, see blue words from line 674 to line 675. Since it is a full sample statistic for the textbook, there is no need for inferential statistics. Thank you again to the reviewers.

Similarly, the description of the critical discourse analysis approach lacks specificity. The authors mention an “improved CDA,” but it is unclear what modifications were made and how they were implemented.

Response: Thank you to the reviewers for their careful review and constructive criticism. To clarify the CDA method, we have explicitly cited three forms of improved CDA in our study, with a particular focus on Fairclough's dialectical-relational approach. We have provided an explanation of the specific features of each approach and highlighted their differences, see blue words from line 222 to line 233, from line 435 to line 474 and from from line 438 to line 526. Thank you again to the reviewers.

The manuscript exhibits significant repetition, particularly in the Results and Discussion sections. Key points, such as the underrepresentation of women and the persistence of gender stereotypes, are reiterated multiple times without adding new insights.

Response: Thank you to the reviewers for their careful review and constructive criticism. We deleted the verbose and repetitive parts of the conclusion and discussion to make the article more concise and clear. Thank you again to the reviewers.

the underrepresentation of women and the persistence of gender stereotypes, are reiterated multiple times without adding new insights. Additionally, the discussion of regression in gender representation is presented in multiple sections but would benefit from a more integrated analysis.

Response: Thank you to the reviewers for their careful review and constructive criticism. We have adjusted section “Underrepresentation of women: gender ideology lags behind the times” to make the analysis more comprehensive,see blue words in line 656. Thank you again to the reviewers.

While the study references existing literature, the engagement with theoretical frameworks and global perspectives on gender representation in textbooks could be strengthened. The manuscript cites key scholars such as Sunderland and Beauvoir but does not fully incorporate theoretical discussions on how gender stereotypes influence students’ career aspirations and self-perception.

Response: Thank you to the reviewers for their careful review and constructive criticism. We removed references to Sunderland et al in conclusion. Thank you again to the reviewers.

Furthermore, drawing on global practices, such as UNESCO’s guidelines for gender-sensitive teaching materials, would provide a stronger foundation for the study’s policy recommendations.

Response: Thank you to the reviewers for their careful review and constructive criticism. In the last paragraphs of the conclusion, We mentioned UNESCO's guidelines, see blue words in from line 686 to line 691. Thank you again to the reviewers.

The manuscript’s readability is hindered by long, complex sentences and occasional grammatical errors. For example, sentences such as “The present study conducted a quantitative and qualitative analysis of numerous core indicators, and found that with the improvement of

---

## [Decision Letter · Decision Letter 1]

15 Jun 2025

Dear Dr. He,

We look forward to receiving your revised manuscript.

Kind regards,

Rafael Galvão de Almeida, PhD.

Academic Editor

PLOS ONE

Journal Requirements:

Reviewers' comments:

Reviewer's Responses to Questions

**Comments to the Author**

Reviewer #2: All comments have been addressed

2. Is the manuscript technically sound, and do the data support the conclusions?

Reviewer #2: Partly

3. Has the statistical analysis been performed appropriately and rigorously?

Reviewer #2: Yes

4. Have the authors made all data underlying the findings in their manuscript fully available?

Reviewer #2: Yes

5. Is the manuscript presented in an intelligible fashion and written in standard English?

Reviewer #2: Yes

Reviewer #2: Review of the article entitled “Progress and Stagnation: A Comparative Study of Gender Representation in Chinese Textbooks for Junior High School between 2001 Edition and 2023 Edition”

The authors have made commendable progress in response to reviewer feedback. The manuscript demonstrates increased clarity in its methodological description and improved engagement with international policy standards. However, further enhancement is still needed in terms of methodological transparency (e.g., coding examples), theoretical integration (particularly educational and philosophical perspectives), and critical discussion of structural influences and limitations. Addressing these remaining issues will significantly strengthen the rigor and scholarly contribution of the manuscript. Here I list my detailed comments:

1. Methodology

First-round comment:

“Despite these strengths, the methodology section requires further elaboration to enhance the study’s transparency and reproducibility. The manuscript does not provide sufficient detail on the coding process, inter-coder reliability, or statistical tests used in the quantitative analysis. Similarly, the description of the critical discourse analysis approach lacks specificity. The authors mention an ‘improved CDA,’ but it is unclear what modifications were made and how they were implemented.”

Authors’ response and assessment:

The authors have taken steps to clarify the coding criteria in Sections 3.2 and 3.3. For instance, in Section 3.2.1 (“Coding criteria”), they specify that “protagonists are the core characters who drive the plot, excluding background characters” (lines 156–158). In Section 3.3.2 (“Data analysis”), the framework for analyzing family roles has been elaborated, adopting a tripartite model of “economic contribution/emotional support/decision-making leadership” (lines 202–205). These additions improve the clarity of the coding scheme.

However, the revision still falls short in two areas. First, the absence of concrete coding examples (e.g., a sample coding table illustrating gender role labeling for a specific text) undermines methodological transparency and limits reproducibility. Second, the application of Critical Discourse Analysis (CDA) remains vague. While Fairclough’s three-dimensional framework is referenced, the operationalization—particularly how “power relations” are identified and coded—has not been sufficiently detailed.

2. Theoretical Engagement

First-round comment:

“While the study references existing literature, the engagement with theoretical frameworks and global perspectives on gender representation in textbooks could be strengthened. The manuscript cites key scholars such as Sunderland and Beauvoir but does not fully incorporate theoretical discussions on how gender stereotypes influence students’ career aspirations and self-perception. Furthermore, drawing on global practices, such as UNESCO’s guidelines for gender-sensitive teaching materials, would provide a stronger foundation for the study’s policy recommendations.”

Authors’ response and assessment:

The authors have incorporated UNESCO’s Gender-Sensitive Textbook Guidelines in the policy recommendations section (lines 686–691), advocating for “balancing gender roles and reducing stereotypical descriptions.” This is a welcome addition.

Nonetheless, one important gap remains. While the manuscript initially referenced Beauvoir, the revised version misses the opportunity to engage in a deeper philosophical dialogue—for example, by connecting the “othering” of female characters in textbooks with Beauvoir’s concept of women as a socially constructed “second nature.” These missed opportunities weaken the theoretical foundation of the study.

3. Practical Implications

First-round comment:

“The study would benefit from a more in-depth exploration of its practical implications for educational policy and practice. While the findings highlight gaps in gender representation, the manuscript does not provide concrete recommendations for textbook authors, policymakers, or educators. Additionally, the omission of texts addressing gender equality is noted, but the manuscript does not fully explore how this might influence students’ perceptions or what corrective actions could be taken.”

Authors’ response and assessment:

The authors have addressed this point by offering specific suggestions in Chapter 6 (Conclusion). For instance, they recommend that textbook authors “design equal social roles, such as increasing the images of female scientists and engineers” (lines 679–680), and that policymakers “develop gender-sensitive teaching material guidelines and clarify gender equality assessment indicators” (lines 688–689). In Chapter 5 (Discussion), they also reflect on the removal of the text “Face Changing,” arguing that it deprives students of the opportunity for critical engagement with gender issues (lines 408–434).

Despite these improvements, the influence of omitted gender-equality texts on students’ cognitive development is still not sufficiently addressed. Incorporating insights from educational psychology could help elucidate the link between textual omission and the reinforcement of gender stereotypes, thus enhancing the empirical and practical relevance of the study’s claims.

4. Acknowledgment of Limitations and Counterarguments

First-round comment:

“Finally, the manuscript would be strengthened by a discussion of counterarguments and study limitations. The analysis attributes gender disparities in textbooks primarily to editorial bias but does not consider alternative explanations, such as broader societal trends or historical constraints on available materials. Additionally, the study focuses on only two editions of Chinese textbooks, limiting its ability to capture long-term trends comprehensively. Acknowledging these limitations and discussing potential biases in the coding and analysis process would enhance the study’s credibility and scholarly contribution.”

Authors’ response and assessment:

The authors now acknowledge in Chapter 6 (Conclusion) that analyzing only two editions restricts the longitudinal scope of the study. They recommend including additional editions in future research (lines 669–675). They also recognize the inherent subjectivity in the coding process and suggest adopting multi-coder strategies and reporting error indicators in future work.

However, a deeper engagement with alternative explanations is still missing. The manuscript briefly notes that “social and cultural traditions may have a greater impact on textbook content than editorial bias,” but this point is not fully developed.

**Do you want your identity to be public for this peer review?** For information about this choice, including consent withdrawal, please see our Privacy Policy

Reviewer #2: No

---

## [Author Response · Author response to Decision Letter 2]

8 Aug 2025

Response to editors

Prior to sharing human research participant data, authors should consult with an ethics committee to ensure data are shared in accordance with participant consent and all applicable local laws.

Response: The images included in the Supplementary Information of this paper are sourced from publicly published textbooks. These images are used exclusively for academic research purposes, in strict compliance with Article 24 of the Copyright Law of the People's Republic of China, which permits the appropriate quotation of previously published works 'for the purpose of introducing, commenting on a certain work, or explaining a certain issue,' provided that such use 'does not affect the normal exploitation of the work and does not unreasonably prejudice the legitimate interests of the copyright owner.' Furthermore, such use 'may be undertaken without permission from the copyright owner and without payment of remuneration, provided that the author’s name (or pseudonym) and the title of the work are indicated.'

Additionally, the use of portraits depicted in these images complies with Article 1,020 of the Civil Code of the People's Republic of China, which stipulates that 'the use of a publicly disclosed portrait within necessary limits for personal study, artistic appreciation, classroom instruction, or scientific research' 'may be made without the consent of the person whose portrait is used.'

Please provide additional details regarding participant consent.

Response: Our research focuses on analyzing publicly published school textbooks, with no copyright disputes involved. It does not involve any human or animal subjects, nor does it involve collecting or harming any living organisms. Therefore, we do not possess any authorization legal documents or ethical statements.

Please include a separate legend for each figure in your manuscript.

Response: We have done. See blue words.

Please upload a copy of Figure 1, 2, 3, 4 and 5 which you refer to in your text on page xx.

Response: We have done. See blue words.

Response:We removed these references:

“[42]Smith, S. L., and A. D. Granados. “Content Patterns and Effects Surrounding Sex-role Stereotyping on Television and Film.” In Media Effects: Advances in Theory and Research, edited by J. Bryant and M. B. Oliver, New York / London : Routledge; 2009. p.342–361.

[43]Midgley, C., DeBues-Stafford, G., Lockwood, P., & Thai, S. She needs to see it to be it: The importance of same-gender athletic role models. Sex Roles. 2021; 85(3): 142–160. DOI: 10.1007/s11199-020-01209-y”

These references were added:

“[19] Zhang, L., Zhang, Y., & Cao, R. "Can we stop cleaning the house and make some food, Mum?": A critical investigation of gender representation in China's English textbooks. Linguistics and Education. 2022; 69, 101058. https://doi.org/10.1016/j.linged.2022.101058

[37] Fairclough, N. Critical Discourse Analysis: The Critical Study of Language. Routledge. 1995, p. 133.

[39] Ueno, C. Patriarchy and Capitalism (Z. Yun & X. Mei, Trans.). Hangzhou: Zhejiang University Press. 2020, p. 48.”

Response to Reviewers

Reviewer #2: Review of the article entitled “Progress and Stagnation: A Comparative Study of Gender Representation in Chinese Textbooks for Junior High School between 2001 Edition and 2023 Edition”

The authors have made commendable progress in response to reviewer feedback. The manuscript demonstrates increased clarity in its methodological description and improved engagement with international policy standards.

Response: Thank you to the reviewer for his/her positive comments.

However, further enhancement is still needed in terms of methodological transparency (e.g., coding examples), theoretical integration (particularly educational and philosophical perspectives), and critical discussion of structural influences and limitations. Addressing these remaining issues will significantly strengthen the rigor and scholarly contribution of the manuscript. Here I list my detailed comments:

1. Methodology

First-round comment:

“Despite these strengths, the methodology section requires further elaboration to enhance the study’s transparency and reproducibility. The manuscript does not provide sufficient detail on the coding process, inter-coder reliability, or statistical tests used in the quantitative analysis. Similarly, the description of the critical discourse analysis approach lacks specificity. The authors mention an ‘improved CDA,’ but it is unclear what modifications were made and how they were implemented.”

Authors’ response and assessment:

The authors have taken steps to clarify the coding criteria in Sections 3.2 and 3.3. For instance, in Section 3.2.1 (“Coding criteria”), they specify that “protagonists are the core characters who drive the plot, excluding background characters” (lines 156–158). In Section 3.3.2 (“Data analysis”), the framework for analyzing family roles has been elaborated, adopting a tripartite model of “economic contribution/emotional support/decision-making leadership” (lines 202–205). These additions improve the clarity of the coding scheme.

However, the revision still falls short in two areas. First, the absence of concrete coding examples (e.g., a sample coding table illustrating gender role labeling for a specific text) undermines methodological transparency and limits reproducibility.

Response: Thank you to the reviewers for their careful review and constructive criticism. See the red words in Table 2 for details of revision. We have added coding cases to table 2 and providing necessary supplementary information.

Second, the application of Critical Discourse Analysis (CDA) remains vague. While Fairclough’s three-dimensional framework is referenced, the operationalization—particularly how “power relations” are identified and coded—has not been sufficiently detailed.

Response: Thank you to the reviewers for their careful review and constructive criticism. We use Fairclough's three-dimensional framework to carefully analyze some of the texts. See red words in the lines 305-310, and in the lines 484-597.

2. Theoretical Engagement

First-round comment:

“While the study references existing literature, the engagement with theoretical frameworks and global perspectives on gender representation in textbooks could be strengthened. The manuscript cites key scholars such as Sunderland and Beauvoir but does not fully incorporate theoretical discussions on how gender stereotypes influence students’ career aspirations and self-perception. Furthermore, drawing on global practices, such as UNESCO’s guidelines for gender-sensitive teaching materials, would provide a stronger foundation for the study’s policy recommendations.”

Authors’ response and assessment:

The authors have incorporated UNESCO’s Gender-Sensitive Textbook Guidelines in the policy recommendations section (lines 686–691), advocating for “balancing gender roles and reducing stereotypical descriptions.” This is a welcome addition.

Nonetheless, one important gap remains. While the manuscript initially referenced Beauvoir, the revised version misses the opportunity to engage in a deeper philosophical dialogue—for example, by connecting the “othering” of female characters in textbooks with Beauvoir’s concept of women as a socially constructed “second nature.” These missed opportunities weaken the theoretical foundation of the study.

Response: Thank you to the reviewers for their careful review and constructive criticism. We further explain the problem of "othering", see red words in the lines 721-728.

3. Practical Implications

First-round comment:

“The study would benefit from a more in-depth exploration of its practical implications for educational policy and practice. While the findings highlight gaps in gender representation, the manuscript does not provide concrete recommendations for textbook authors, policymakers, or educators. Additionally, the omission of texts addressing gender equality is noted, but the manuscript does not fully explore how this might influence students’ perceptions or what corrective actions could be taken.”

Authors’ response and assessment:

The authors have addressed this point by offering specific suggestions in Chapter 6 (Conclusion). For instance, they recommend that textbook authors “design equal social roles, such as increasing the images of female scientists and engineers” (lines 679–680), and that policymakers “develop gender-sensitive teaching material guidelines and clarify gender equality assessment indicators” (lines 688–689). In Chapter 5 (Discussion), they also reflect on the removal of the text “Face-Changing,” arguing that it deprives students of the opportunity for critical engagement with gender issues (lines 408–434).

Despite these improvements, the influence of omitted gender-equality texts on students’ cognitive development is still not sufficiently addressed. Incorporating insights from educational psychology could help elucidate the link between textual omission and the reinforcement of gender stereotypes, thus enhancing the empirical and practical relevance of the study’s claims.

Response: Thank you to the reviewers for their careful review and constructive criticism. We use educational psychology theory to further illustrate the impact of gender inequality on students' psychological development. See red words in the lines 31-33, and in the lines 695-700.

4. Acknowledgment of Limitations and Counterarguments

First-round comment:

“Finally, the manuscript would be strengthened by a discussion of counterarguments and study limitations. The analysis attributes gender disparities in textbooks primarily to editorial bias but does not consider alternative explanations, such as broader societal trends or historical constraints on available materials. Additionally, the study focuses on only two editions of Chinese textbooks, limiting its ability to capture long-term trends comprehensively. Acknowledging these limitations and discussing potential biases in the coding and analysis process would enhance the study’s credibility and scholarly contribution.”

Authors’ response and assessment:

The authors now acknowledge in Chapter 6 (Conclusion) that analyzing only two editions restricts the longitudinal scope of the study. They recommend including additional editions in future research (lines 669–675). They also recognize the inherent subjectivity in the coding process and suggest adopting multi-coder strategies and reporting error indicators in future work.

However, a deeper engagement with alternative explanations is still missing. The manuscript briefly notes that “social and cultural traditions may have a greater impact on textbook content than editorial bias,” but this point is not fully developed.

Response: Thank you to the reviewers for their careful review and constructive criticism.We have made an in-depth discussion of gender issues in Chinese culture, as shown the red words on lines 73-88,795-800, and 818-821.

Finally, We would like to express our sincere thanks to the editors and reviewers

---

## [Decision Letter · Decision Letter 2]

2 Sep 2025

Progress and Stagnation: A Comparative Study of Gender Representation in Chinese Textbooks for Junior High School between 2001 Edition and 2023 Edition

PLOS ONE

Dear Dr. He,

Thank you for submitting your manuscript to PLOS ONE. After careful consideration, we feel that it has merit but does not fully meet PLOS ONE’s publication criteria as it currently stands. Therefore, we invite you to submit a revised version of the manuscript that addresses the points raised during the review process.

We look forward to receiving your revised manuscript.

Kind regards,

Zahra Al-Khateeb, Ph.D

Staff Editor

PLOS ONE

On behalf of Rafael Galvão de Almeida, PhD. 

Journal Requirements:

Comments from the editorial office: 

PLOS ONE publication criteria require that the data presented in the manuscript must support the conclusions drawn, and submissions will be rejected if the interpretation of results is unjustified or inappropriate, so authors should avoid overstating their conclusions. We noted potentially overstated and/or unclear statements in the introduction and discussion sections that require revision. Please ensure all statements within the submission are appropriately supported by citations or the results presented in the manuscript.

For example:

Line 73-86: "China is an ancient country with the world-renowned Confucian culture. Confucianism has been traditionally valued in China to sustain a patriarchal society. Influenced by Confucian values, men are seen as representatives and inheritors of the family, while women are expected to marry into other families and continue their bloodline from the family they marry into. Therefore, men are encouraged to pursue careers, while women are required to take care of their husbands and children, look after their parents-in-law, and do laundry and cooking. The standard for measuring a woman's value is to be a "virtuous wife" to her husband's family. Housework is not recognized by society and lacks economic value. Men are authorities in the family domain, and their power is further strengthened and consolidated through gender division of labor in social life. This shows the "two-dimensional patriarchal system functioning in social and familial spheres" rooted in China [19]. Therefore, in ancient China, women were marginalized by society and their behavior and status were generally dominated by men"

Line 93-97: "Since 2000, the status of Chinese women has seen tremendous improvement. In universities, the proportion of female students exceeds 50%, and in hospitals, the percentage of female doctors is over 46%. Furthermore, women account for 35% to 38% of corporate board directors and supervisors, and constitute more than 50% of grassroots neighborhood committee members."

Line 554: "Gogchild" (This is a typo, as it should be "Dogchild")

Line 666-667: "however, there are few characters in Chinese textbooks that counter stereotypes."

Line 731-732: "The patriarchal gender norm holds that women should be the other in the eyes of men and obey the will of men"

We noted this concern throughout the manuscript, with the above serving as a few examples. Please carefully review the manuscript and revise the introduction, results and discussion sections to ensure that everything is clearly stated, all claims are well-supported by the data, and the tone is neutral.

Additional Editor Comments:

Kindly review the comments provided by the Editorial Office below.

Reviewer's Responses to Questions

**Comments to the Author**

Reviewer #2: All comments have been addressed

2. Is the manuscript technically sound, and do the data support the conclusions?

Reviewer #2: Yes

3. Has the statistical analysis been performed appropriately and rigorously?

Reviewer #2: Yes

4. Have the authors made all data underlying the findings in their manuscript fully available?

Reviewer #2: Yes

5. Is the manuscript presented in an intelligible fashion and written in standard English?

Reviewer #2: Yes

Reviewer #2: Dear editor,

Thank you for inviting me to review this manuscript for the third time. I truly appreciate your rigorous and thoughtful approach throughout this review process.

I have read the manuscript carefully and would also like to commend the authors for their careful revisions. The revised manuscript has adequately addressed all the concerns and suggestions I raised in my previous reviews. Therefore, I am pleased to endorse the publication of this article and would like to congratulate the authors in advance on its forthcoming acceptance and eventual publication.

Xiaoming

**Do you want your identity to be public for this peer review?** For information about this choice, including consent withdrawal, please see our Privacy Policy

Reviewer #2: No

---

## [Author Response · Author response to Decision Letter 3]

23 Sep 2025

Response to Editors and Reviewers

Response to editors

Response: We checked our reference list, deleted two references and adjusted some references, as shown in blue words in reference list.

PLOS ONE publication criteria require that the data presented in the manuscript must support the conclusions drawn, and submissions will be rejected if the interpretation of results is unjustified or inappropriate, so authors should avoid overstating their conclusions. We noted potentially overstated and/or unclear statements in the introduction and discussion sections that require revision. Please ensure all statements within the submission are appropriately supported by citations or the results presented in the manuscript.

Response: Thank you to the editors for their careful review and constructive criticism. The paper has been carefully proofread and a lot of changes have been made to the problems mentioned by the editor. See the blue text in paper.

Please ensure all statements within the submission are appropriately supported by citations or the results presented in the manuscript.

For example:Line 73-86: "China is an ancient country with the world-renowned Confucian culture. Confucianism has been traditionally valued in China to sustain a patriarchal society. Influenced by Confucian values, men are seen as representatives and inheritors of the family, while women are expected to marry into other families and continue their bloodline from the family they marry into. Therefore, men are encouraged to pursue careers, while women are required to take care of their husbands and children, look after their parents-in-law, and do laundry and cooking. The standard for measuring a woman's value is to be a "virtuous wife" to her husband's family. Housework is not recognized by society and lacks economic value. Men are authorities in the family domain, and their power is further strengthened and consolidated through gender division of labor in social life. This shows the "two-dimensional patriarchal system functioning in social and familial spheres" rooted in China [19]. Therefore, in ancient China, women were marginalized by society and their behavior and status were generally dominated by men"  

Response: Thank you to the editors for their careful review and constructive criticism�I've already deleted that part.

"Since 2000, the status of Chinese women has seen tremendous improvement. In universities, the proportion of female students exceeds 50%, and in hospitals, the percentage of female doctors is over 46%. Furthermore, women account for 35% to 38% of corporate board directors and supervisors, and constitute more than 50% of grassroots neighborhood committee members."

Response: Thank you to the editors for their careful review and constructive criticism, I've added two references to this section, see blue words in line 78-83.

"Gogchild" (This is a typo, as it should be "Dogchild")

Response: Thank you to the editors for their careful review and constructive criticism, we have corrected the error and therefore apologize.

"however, there are few characters in Chinese textbooks that counter stereotypes."

Response: Thank you to the editors for their careful review and constructive criticism�I've already deleted that part.

"The patriarchal gender norm holds that women should be the other in the eyes of men and obey the will of men".

Response: Thank you to the editors for their careful review and constructive criticism�I've already deleted that part.

Response to Reviewers

Reviewer #2: Thank you for inviting me to review this manuscript for the third time. I truly appreciate your rigorous and thoughtful approach throughout this review process.

I have read the manuscript carefully and would also like to commend the authors for their careful revisions. The revised manuscript has adequately addressed all the concerns and suggestions I raised in my previous reviews. Therefore, I am pleased to endorse the publication of this article and would like to congratulate the authors in advance on its forthcoming acceptance and eventual publication.

Xiaoming

Response: Thank you to the reviewer for his/her positive comments.

Finally, We would like to express our sincere thanks to the editors and reviewers.

---

## [Editor Report · Decision Letter 3]

28 Sep 2025

Progress and Stagnation: A Comparative Study of Gender Representation in Chinese Language textbooks for Junior High School between 2001 Edition and 2023 Edition

PONE-D-25-05629R3

Dear Dr. He,

We’re pleased to inform you that your manuscript has been judged scientifically suitable for publication and will be formally accepted for publication once it meets all outstanding technical requirements.

Kind regards,

Rafael Galvão de Almeida, PhD.

Academic Editor

PLOS ONE
---

## [Editor Report · Acceptance letter]

PONE-D-25-05629R3

PLOS ONE

Dear Dr. He,

I'm pleased to inform you that your manuscript has been deemed suitable for publication in PLOS ONE. Congratulations! Your manuscript is now being handed over to our production team.

Kind regards,

on behalf of

Dr. Rafael Galvão de Almeida

Academic Editor

PLOS ONE